# A Novel Polysaccharide from Blackened Jujube: Structural Characterization and Immunoactivity

**DOI:** 10.3390/foods14142531

**Published:** 2025-07-19

**Authors:** Meng Meng, Fang Ning, Xiaoyang He, Huihui Li, Yinyin Feng, Yanlong Qi, Huiqing Sun

**Affiliations:** 1State Key Laboratory of Food Nutrition and Safety, Ministry of Education, College of Food Science and Engineering, Tianjin University of Science and Technology, No. 29, 13th Avenue, Tianjin Economy Technological Development Area, Tianjin 300457, China; mengmeng@tust.edu.cn (M.M.); ningfang12022@163.com (F.N.); bunny_he@126.com (X.H.); lhh010309@163.com (H.L.); fengyin-yin2022@163.com (Y.F.); 2Institute of Agricultural Products Processing, Xinjiang Academy of Agricultural Sciences, Xinjiang Key Laboratory of Processing and Preservation of Agricultural Products, No. 403 Nanchang Road, Urumqi 830091, China

**Keywords:** blackened jujube, polysaccharide, structural characterization, immunoregulatory activity

## Abstract

Previously, research adopted an ultrasound-assisted extraction method to isolate crude polysaccharide from blackened jujube, followed by preliminary structural identification of the purified polysaccharide (BJP). This manuscript analyzed the accurate structure and immunomodulatory activity of BJP. Further structural identification indicated that BJP was mainly composed of →3)-α-L-Ara*f*-(1→, →3,5)-α-L-Ara*f*-(1→, →3)-β-D-Gal*p*A-(1→, →2,4)-β-D-Gal*p*-(1→, →4)-β-D-Gal*p*A-(1→, →3)-α-L-Rha*p*-(1→ and →3,4)-α-L-Rha*p*-(1→. The immunomodulatory effects of BJP were examined using a mouse model with immunosuppression induced by cyclophosphamide. The findings suggested that BJP could relieve the condition of immunosuppressed mice. BJP could inhibit decreases in the body weight and organ index of mice, and HE staining showed that BJP could alleviate the harm to spleen and thymus tissues. BJP enhanced the secretion of interferon-γ (IFN-γ), tumor necrosis factor-α (TNF-α), interleukin-2 (IL-2), interleukin-6 (IL-6), immunoglobulin A (IgA), and immunoglobulin G (IgG) in serum. It also reduced liver oxidative stress by increasing superoxide dismutase (SOD), catalase (CAT), and glutathione (GSH) activities, while lowering malondialdehyde (MDA) levels. Moreover, BJP contributed to the maintenance of gut homeostasis by stimulating the generation of short-chain fatty acids in the cecal contents. The study aims to establish a solid basis for the comprehensive development of blackened jujube and furnish a theoretical framework for its polysaccharides’ role in immune modulation.

## 1. Introduction

The production of blackened jujube involves subjecting the fruit to controlled conditions of high temperature and humidity. Non-enzymatic browning, Maillard reaction, and macromolecular degradation are the main reactions in the melanization process [1]. Compared to red jujube, blackened jujube showed higher nutritional value after fermentation. As an important active ingredient, jujube polysaccharide has multiple biological activities such as immune enhancement [2], antioxidation [3], anti-tumor properties [4], intestinal health maintenance, and liver protection [5]. To date, most research on blackened jujube has centered on the changes that occur during the blackening process, with fewer studies investigating the polysaccharides and their biological activities. Yuan et al. [6] purified five polysaccharide components from blackened jujube and studied their structural characteristics and antioxidant activities. The five purified polysaccharides from blackened jujube varied significantly in chemical composition, molecular weight, types of monosaccharides, and infrared spectra. Furthermore, they also varied in their in vitro free radical scavenging abilities and protection against HUVEC injury triggered by H_2_O_2_. Liu et al. [7] obtained the purified polysaccharide identified as BJP-4, with its molecular weight determined to be 1.24 × 10^5^ Da, primarily comprised of the following sugar residues: →4)-α-L-Gal*p*A(1→, →5)-α-L-Ara*f*-(1→ and →4)-β-D-Gal*p*(1→). In addition, it was found that BJP-4 could effectively alleviate the symptoms of colitis in mice, including reducing the disease activity index, improving colon length, and healing pathological damage. The mitigation mechanisms of BJP-4 on colitis mice included regulating the composition of intestinal flora, inhibiting protein expression in the TLR4/MyD88/NF-κB/NLRP3 signaling pathway, maintaining the balance of inflammatory cytokine levels, regulating oxidative stress, strengthening the intestinal epithelial barrier’s integrity, and facilitating the production of short-chain fatty acids.

When confronted with bacteria, viruses, and other pathogens, the body’s immune defense system can quickly recognize these foreign stimuli and start the corresponding response to remove pathogens, prevent the spread of infection, and then maintain the stability and health of the body. Two principal aspects make up the immune system: innate immunity and acquired immunity. As a natural defense system of the body, innate immunity can trigger a series of complex immune reactions in the innate immune cell population (including monocytes, macrophages, neutrophils, natural killer cells, etc.) when pathogens invade, and then stimulate the cell response to produce corresponding cytokines, showing the ability of innate immune memory [8]. The immune response that develops in the body after encountering a specific pathogen is known as acquired immunity, which consists of humoral and cellular components. Humoral immunity primarily recognizes and eliminates invading pathogens through antibodies produced by B lymphocytes, while cellular immunity relies on T lymphocytes directly attacking infected cells [9]. Polysaccharides fulfill their immunomodulatory functions by binding to a variety of pattern recognition receptors (PRRs) on cellular surfaces. These receptors include the Toll-like receptor family, scavenger receptor family, Dectin-1 receptor, and complement receptor [10,11,12]. After activation, polysaccharide activates the signal channels downstream of these receptors, activate a variety of immune cells, and induce the synthesis of immune cytokines, immunoglobulins, and antibodies, thus strengthening the body’s immune protective functions [13,14,15].

Given their low toxicity and strong biological activities, natural polysaccharides are gaining attention as a promising research area within immune regulation. Research on the biological effects of polysaccharides extracted from blackened jujube is currently quite restricted. Previous studies have optimized the extraction process of blackened jujube through ultrasonic-assisted extraction. The polysaccharide in blackened jujube showed strong antioxidant activity by scavenging hydroxyl radicals, DPPH radicals, and superoxide anions. The purification process to obtain blackened jujube polysaccharide (BJP) involved removing proteins, conducting dialysis to remove impurities, and using a Sephadex G-100 gel column. The molecular weight of BJP was 1.42 × 10^6^ Da. It was found to be mainly composed of rhamnose (Rha), arabinose (Ara), galactose (Gal), and galacturonic acid (Gal-UA) according to the ion chromatography results, with molar ratios of 0.77:2.47:1:1.46 for each monosaccharide, respectively [16]. Considering the established relationship between immune function and antioxidant and anti-inflammatory capabilities, further exploration of the immunomodulatory effects of blackened jujube polysaccharide may provide new scientific evidence for their broader applications. This manuscript characterized the accurate structure of blackened jujube polysaccharides and investigated their immunomodulatory activity, laying the foundation for the comprehensive utilization of blackened jujube polysaccharide. It could provide theoretical support for blackened jujube polysaccharide to be potential immunomodulators.

## 2. Materials and Methods

### 2.1. Materials and Reagents

Blackened jujube was provided by Xinjiang Yuweixian Agricultural Science and Technology Center, Xinjiang, China. Cyclophosphamide was bought from Shanghai Yuanye Biotechnology Co., LTD, Shanghai, China. Levamisole hydrochloride was obtained from Shanghai Yien Chemical Co., LTD, Shanghai, China. Trifluoroacetic acid, inositol hexacetate, and short-chain fatty acid standard were sourced from Shanghai McLean Biochemical Technology Co., LTD, Shanghai, China. Methanol, methylene chloride, and pyridine were procured from Sinopharm Group Chemical Reagent Co., LTD, Beijing, China. Other analytical-grade chemical reagents were purchased from Tianjin Jiangtian Chemical Co., LTD, Tianjin, China.

### 2.2. Preparation of BJP

Referring to the method in the literature [16], the crude polysaccharide of black jujube was extracted in three repetitions using ultrasonic extraction equipment (BILON10-250C, Shanghai Bilon Instrument Manufacturing Co., LTD, Shanghai, China). The purified blackened jujube polysaccharide was obtained by removing proteins with Sevag reagent, removing impurities with 3500 Da dialysis bags, and passing it through Sephadex G-100 chromatographic columns ( Beijing Solaibao Technology Co., Ltd., Beijing, China), and it was named BJP.

### 2.3. Structure Analysis of BJP

#### 2.3.1. Methylation Analysis

Referring to Hong et al. [17], 15 mg of dried BJP was reduced. The reduced BJP was combined with 25 mg NaH and 2 mL DMSO, and the resulting mixture underwent ultrasonic treatment in an ice bath protected by N_2_ for 2 h. Subsequently, 2 mL iodomethane was incorporated, followed by N_2_ purge. The ultrasonic reaction was carried out in an ice bath for 12 h. Afterwards, the solution was dried using N_2_. All of the above reactions were performed in the dark and repeated until infrared spectroscopy (IS50, Thermo Nicolet Corporation, Madison, Wisconsin, USA) confirmed successful methylation. The reaction was then quenched with the addition of 0.5 mL of distilled water, and a matching volume of dichloromethane was employed to separate the organic fraction. The sample was then dried using N_2_. Next, the dried sample was treated with 2 mL of 2 mol/L trifluoroacetic acid, and the reaction was conducted at 110 °C for 3 h. After cooling at the end of the reaction, N_2_ was used to dry the solution. Then, 2 mL of methanol was mixed in to dissolve the mixture entirely, and the mixture was dried using N_2_ again. This process was repeated four times. Subsequently, 2 mL of aqueous solution with 25 mg of sodium borohydride was introduced. After 2 h of reaction, the pH was adjusted to approximately 5 using 0.2 mol/L acetic acid, and the solvent was dried with N_2_. Next, methanol (2 mL) and acetic acid (50 μL) were mixed five times, followed by N_2_ blow-drying each time. The procedure involved introducing 1 mL each of pyridine and acetic anhydride to the blow-dried sample, which was subsequently placed in an oil bath at 100 °C for 1 h. The sample was dried using N_2_ after it had cooled down. In order to remove residual substances, 3 mL of methanol was applied to the blow-dried sample, and the steps of methanol washing and blow-drying were carried out three times in a row. Finally, the methylated PMAAS was dissolved in dichloromethane, filtered through organic membranes, analyzed by GC-MS (GCMS7890B-7000C, Agilent Technologies Inc., Santa Clara, CA, USA), and compared to a database, which in turn determined the type of methylated sugar residues.

#### 2.3.2. Nuclear Magnetic Analysis

After dissolving 50 mg of BJP in 0.6 mL D_2_O, the solution was carefully poured into a nuclear magnetic tube. The nuclear magnetic resonance (NMR) spectrometer (AVIII400M, Bruker Corporation, Billerica, Massachusetts, USA) was used to obtain the ^1^H-NMR, ^13^C-NMR, heteronuclear single quantum coherence spectroscopy (HSQC), and correlation spectroscopy (COSY) spectra of the solution.

#### 2.3.3. Congo Red Analysis

At room temperature, 0.5 mg/mL of BJP solution was blended with 50 µmol/L of Congo red solution. Then, the final NaOH concentrations were adjusted to 0, 0.05, 0.10, 0.15, 0.20, 0.30, and 0.40 mol/L by adding a 1 mol/L NaOH solution to the mixture. After 10 min of reaction, the solutions were scanned with a UV spectrophotometer (UV-1600, Shanghai Meipuda Instrument Co., LTD, Shanghai, China) between 400 and 600 nm, and the wavelength with the maximum absorption was recorded. A trend plot was constructed using the final NaOH concentration as the X-axis and the maximum absorption wavelength as the Y-axis.

### 2.4. In Vivo Immune Activity

#### 2.4.1. Animal and Experimental Design

A total of forty-eight SPF-grade male Kunming mice, with an average body weight of 20 ± 2 g and an age of 4 ± 2 weeks, were provided by Spife (Beijing) Biotechnology Co., Ltd, Beijing, China. The animal experiments were performed according to the laboratory animal care and use guidelines set by the Tianjin International Biomedical Joint Research Institute.

KM mice were allocated into 6 groups at random: a normal control group (NC group), a model group (MC group), a blackened jujube polysaccharide low-dose group (BJP-L), a blackened jujube polysaccharide medium-dose group (BJP-M), a blackened jujube polysaccharide high-dose group (BJP-H), and a positive group (PC group). Following 7 days of adaptive feeding, all groups except the normal control were administered 80 mg/kg bw·d cyclophosphamide (CTX) via intraperitoneal injection for a week to induce an immunosuppressive mouse model. The BJP-L, BJP-M, and BJP-H group mice were given BJP suspensions through oral gavage at respective doses of 200, 400, and 800 mg/kg bw·d. Mice in PC group were given levamisole hydrochloride 40 mg/kg bw·d. Mice in the NC and MC groups were given the same amount of normal saline daily for 16 days.

#### 2.4.2. Body Weight and Organ Index

During the experiment, the weight gain or loss and feeding behavior of the mice were closely observed and recorded. After withholding food and water from the mice for 12 h, their final body weights were measured and recorded as the experiment concluded. Blood collection preceded euthanasia, after which the thymus and spleen were swiftly dissected and collected. Blood stains on the organs were washed with normal saline, excess tissue was removed, and the water from the organs was absorbed using filter paper before weighing. The calculation of the organ index was performed based on Formula (1).
(1)$$OI=\frac{OW}{BW}$$ where *OI* is the organ index (mg/g), *OW* is the organ mass (mg), and *BW* is the body weight (g).

#### 2.4.3. Histopathological Analysis

After being fixed with 4% paraformaldehyde, the thymus and spleen from the mice were processed through an automatic tissue processing system for pathology (Leica Microsystems (Shanghai) Co., LTD., Shanghai, China) and then embedded into wax blocks. Subsequently, the tissue slices were cut into 5 μm slices by a microtome, and the tissue slice samples were finally prepared by hematoxylin–eosin staining, decolorization, transparency, and sealing. The prepared section samples were then observed and analyzed under an optical microscope (DM4000, Germany Leica Instrument Co., LTD, Wetzlar, Germany).

#### 2.4.4. Determination of Inflammatory Factors in Mouse Serum

The serum levels of interferon-γ (IFN-γ), tumor necrosis factor-α (TNF-α), interleukin-2 (IL-2), and interleukin-6 (IL-6) in mice were measured using commercial Mouse ELISA kits (SEKM-0031, SEKM-0034, SEKM-0004, and SEKM-0007; Beijing Solarbio Science & Technology Co., Ltd., Beijing, China).

#### 2.4.5. Determination of Immunoglobulin in Mouse Serum

The levels of immunoglobulins A and immunoglobulins G (IgA and IgG) in the serum of mice were detected using commercial Mouse ELISA kits (SEKM-0094, SEKM-0098; Beijing Solarbio Science & Technology Co., Ltd., Beijing, China).

#### 2.4.6. Determination of Antioxidant Capacity of Liver

The levels of catalase (CAT), superoxide dismutase (SOD), reduced glutathione (GSH), and malondialdehyde (MDA) in the liver of mice were detected using commercial mouse enzyme-linked immunosorbent assay kits (BC0205, BC5165, BC1175, BC0025, Beijing Solarbio Science & Technology Co., Ltd., Beijing, China).

#### 2.4.7. Determination of Short-Chain Fatty Acids

Adapting the method described by Wang et al. [18], with several alterations, 25 mg of cecal contents were dissolved with 250 μL of saturated sodium chloride solution and homogenized after 30 min. The process involved adding 20 μL of sulfuric acid to the solution to vortex-mix it, and then 300 μL of ether was added for extraction. Centrifugation of the mixture at 4 °C for 15 min at 12,000 g yielded the supernatant. This supernatant was subsequently dried using anhydrous sodium sulfate for 10 min and then extracted once more under the same centrifugal parameters. The required sample was extracted after passing it through an organic membrane filtration. The samples were examined using a gas chromatograph (GC-010pro, Shimadzu Instruments (Suzhou) Co., LTD., Suzhou, China) that included a hydrogen flame ionization detector and utilized an HP-INNOWAX capillary column (specification: 30 m × 0.25 mm × 0.25 μm).

#### 2.4.8. Statistical Analysis

The data were presented as mean ± SD. Statistical significance was determined through a univariate analysis of variance (ANOVA), with subsequent Tukey’s tests carried out using SPSS 23.0 (*p* < 0.05). Charts were produced using Origin 2019.

## 3. Results

### 3.1. Structural Characterization of BJP

#### 3.1.1. Methylation Analysis

In order to verify the complete methylation of BJP, FT-IR was used for detection. Compared to the infrared spectrum before BJP methylation (Figure 1), the absorption peak of the infrared spectrum after BJP methylation became a more significant narrow peak at 3435 cm^-1^, while the absorption peak at around 2978 cm^-1^ was obviously wider, indicating that the methylation of BJP was complete. Further tests were conducted. GC-MS was used to detect sugar residues to roughly infer the type of glycosidic bonds in the polysaccharide. According to the GC-MS analysis, the derivatives produced by BJP degradation were analyzed and summarized (Table 1). And the following seven sugar residues were 1,3,4-tri-O-Ac_3_-2,5-di-O-Me_2_-arabinitol, 1,3,4,5-tetra-O-Ac_4_-2-O-Me-arabinitol, 1,3,5-tri-O-Ac_3_-2,4,6-di-O-Me_3_-galactitol, 1,2,4,5-tetra-O-Ac_4_-3,6-di-O-Me_2_-galactitol, 1,4,5-tri-O-Ac_3_-2,3,6-di-O-Me_3_-galactitol, 1,3,5-tri-O-Ac_3_-6-deoxy-2,4-di-O-Me_2_-mannitol, and 1,3,4,5-tetra-O-Ac_4_-6-deoxy-2-O-Me-mannitol, with percents of 0.63:3.56:1.08:1.71:1.00:0.67:0.64. The core structure was analyzed by the presence of the following glycosidic bonds of →3)-Araf-(1→, →3,5)-Araf-(1→, →3)-GalpA-(1→, →2,4)-Galp-(1→, →4)-GalpA-(1→, →3)-Rhap-(1→, and →3,4)-Rhap-(1→.

#### 3.1.2. Nuclear Magnetic Analysis

The conformation and connectivity of the glycosidic bonds were further determined from the ^1^HNMR, ^13^CNMR, COSY, and HSQC spectra in the NMR spectra. The anomeric proton shows signals between 4.3 and 5.9 ppm in ^1^H NMR spectra, whereas the anomeric carbon exhibits signals ranging from 90 to 110 ppm in ^13^C NMR spectra. The signals for the α-anomeric proton and carbon in ^1^H NMR and ^13^C NMR fall within the ranges of 4.9–5.3 ppm and 98-103 ppm, respectively, while β-configuration anomeric proton and anomeric carbon signals appear at 4.3–4.9 ppm and 103–106 ppm [19].

As shown in Figure 2A,B, based on the signal of anomeric proton and anomeric carbon in ^1^H NMR and ^13^C NMR, it was inferred that BJP contained α-configuration and β-configuration glycosidic bonds. Additionally, the ^13^C NMR spectra revealed a chemical shift at 170.9 ppm, pointing to the presence of uronic acid as part of an acidic polysaccharide in BJP, which corresponded to the outcomes from prior BJP analyses. Based on the aforementioned results and in conjunction with NMR analysis, the primary focus was on Rha, Ara, Gal, and Gal-UA in BJP, while other monosaccharides exhibited weak signals due to their low concentrations. Therefore, based on the nuclear magnetic resonance results, the analysis of these four monosaccharides was the focus of the subsequent study. Combined with ^1^H spectrum (Figure 2A) and ^13^C spectrum (Figure 2B), the C1/H1 chemical shifts in BJP were observed to be δ108.36/5.38 ppm, δ109.38/5.13 ppm, δ107.16/5.23 ppm, δ107.23/5.1 ppm, δ107.52/5.07 ppm, δ98.52/5.18 ppm, and δ99.15/5.05 ppm, respectively. The glycosidic bonds matched the sugar residues A, B, C, D, E, F, and G sequentially, as seen in the HSQC spectra (Figure 2E) and COSY spectra (Figure 2C,D). The COSY spectrum was used to attribute the signals of adjacent hydrogen protons in the sugar residues. The chemical shifts in the H1-H6 signals in the A-F sugar residues were determined according to the COSY spectrum (Figure 2C,D). The chemical shift relationship between hydrogen atoms and their corresponding carbon atoms in the HSQC spectrum (Figure 2E), combined with the information from the ^13^C NMR spectrum (Figure 2B), was used to complete the attribution of each carbon signal. These analyses indicated that the glycosidic bonds belonged to →3)-α-L-Araf-(1→, →3,5)-α-L-Araf-(1→, →3)-β-D-GalpA-(1→, →2,4)-β-D-Galp-(1→, →4)-β-D-GalpA-(1→, →3)-α-L-Rhap-(1→ and →3,4)-α-L-Rhap-(1→. The attribution of C and H to the seven sugar residues was summarized in Table 2.

#### 3.1.3. Congo Red Analysis

According to the Congo red experiment (Figure 3), at NaOH concentrations from 0 to 0.1 M, in relation to the pure Congo red solution, the ultraviolet absorption spectrum of the mixed solution of BJP and Congo red showed a redshift phenomenon, indicating that a complex was formed between BJP and Congo red, and it was speculated that BJP might contain a regular spiral structure. However, when the concentration of NaOH continued to rise and entered the strong alkaline condition, the hydrogen bonds between molecules were weakened and even destroyed, resulting in the gradual collapse of the spiral structure of the polysaccharide, and finally, the ultraviolet absorption wavelength was decreased. These results corroborated a prior study [20], confirming that BJP has a triple-helix structure.

### 3.2. In Vivo Immunity

#### 3.2.1. Measurement of Body Weight and Organ Index

The indices of mice were observed every day during the experiment. After the experiment, the data on the body weight and organs of mice in each group were collected and calculated according to SPF animal experiment specifications. As shown in Figure 4A, after 4 days of intraperitoneal injection of CTX, the MC group mice displayed significant weight loss, hair thinning, and poor health status in comparison to the NC group. These findings pointed to the successful establishment of an immunocompromised mouse model. In comparison to the MC group, the PC and BJP groups demonstrated a marked improvement in the health of the mice, with a notable suppression of weight loss observed (*p* < 0.05). Within the BJP dose cohort, the BJP-H group’s mice showed a recovery in weight that was statistically indistinguishable from the NC group (*p* < 0.05).

The thymus and spleen are integral to the body’s immune defenses, and their developmental progress is an important gauge of immune response effectiveness. According to Figure 4B,C, mice in the CTX-induced MC group exhibited markedly decreased spleen and thymus indices relative to the NC group (*p* < 0.05). Relative to the MC group, spleen and thymus indices exhibited a dose-dependent increase following BJP gavage, with BJP-H demonstrating the most significant therapeutic benefit. The spleen and thymus indices from the experiment matched those of the PC group (*p* < 0.05).

#### 3.2.2. Histopathological Analysis

The structural integrity and cell composition of the spleen and thymus were further observed by optical microscopy. The splenic tissue morphology was presented in Figure 4D. In the NC group, the splenic white pulp structure was complete, the boundary was clear, and the splenic corpuscle shape was obvious. In contrast, the MC group exhibited obvious pathological changes: the white pulp’s tissue structure was damaged, the distribution of lymphocytes was sparse, the boundary between the white pulp and the red pulp became blurred, and the periarterial lymphatic sheath structure showed signs of loosening. Upon intragastric delivery of BJP, the delineation between the red pulp and white pulp in the spleen improved significantly, the quantity of splenic corpuscles increased, and the white pulp area broadened with higher doses. The basic structure of the white pulp and red pulp of the spleen in the BJP-H group was close to that in the NC group, and the original shape of the spleen tissue gradually recovered. Figure 4E of thymus morphology showed that in the NC group, the proportion of medulla and cortex in the thymus was moderate, and lymphocytes were closely arranged. The cortical area was reduced, lymphocytes were lower, and the boundaries were less distinct in the MC group compared to the NC group. After BJP intragastric treatment, the thymic cortex area of each dose group increased, the boundary was clear, and the lymphocyte arrangement recovered. The proportion of the BJP-H group was close to that of the NC group, and the thymus structure gradually became normal. These findings demonstrated that BJP was capable of alleviating CTX-caused injury to the spleen and thymus tissues and enhancing the recovery of immune organ architecture.

#### 3.2.3. Determination of Inflammatory Factors in Serum

When the pathogen invades, the immune system can activate various immune cells, and the increase in inflammatory factors secreted by immune cells can markedly enhance the body’s immune capability [21]. Inflammatory factors such as TNF-α, INF-γ, IL-6, and IL-2 are important components of cell proliferation. In addition, IL-6 aids in the immune response by increasing antibody production, which results in anti-tumor and anti-inflammatory benefits [22]. The inflammatory factors IL-2, TNF-α, and IFN-γ are capable of boosting the growth and activation of T cells. Additionally, they contribute to the development of memory T cells, thus ensuring long-lasting immune defense for the body [23]. The results presented in Figure 5A–D indicated that the serum contents of TNF-α, INF-γ, IL-2, and IL-6 in MC group mice were significantly reduced relative to the NC group (*p* < 0.05). The results showed that the secretion of inflammatory factors in mice induced by CTX was reduced. In contrast to the MC group, the PC group and the BJP group showed a reversal of the CTX-induced changes in inflammatory factors levels in mice (*p* < 0.05). The BJP dose groups experienced a dose-dependent rise in inflammatory factors levels, with BJP-H exhibiting the most promising therapeutic outcomes, comparable to the PC group (*p* < 0.05). The outcomes revealed that BJP could effectively enhance the secretion of inflammatory factors in immunosuppressed mice, thus improving the immunosuppressed state and enhancing immune function.

#### 3.2.4. Determination of Immunoglobulin in Serum

Immunoglobulins are proteins with antibody activity that are produced and secreted by plasma cells, forming a crucial component of the humoral immune system. Immunoglobulins are vital components in resisting infections and ensuring the immune system remains balanced and stable [24]. Among the various immunoglobulins in the blood, IgG is present in the highest concentration and is essential for antibody-based immune defense [25].

As a key immunoglobulin, IgA is essential for humoral immunity and also makes a critical contribution to the body’s defense against harmful pathogens [26]. As shown in Figure 5E,F, IgA and IgG secretion in the serum of MC group mice was notably lower than that in the NC group (*p* < 0.05), suggesting an impact on humoral immune function. Compared to the NC group, the contents of IgA and IgG gradually recovered after BJP. BJP-M and BJP-H showed similar therapeutic outcomes, and neither differed significantly from the PC group (*p* < 0.05). The results indicated that the polysaccharides of BJP could promote the production of immunoglobulin in immunosuppressed mice and improve the humoral immune function.

#### 3.2.5. Determination of Antioxidant Capacity of Liver Tissue

Oxidative stress is closely related to immune inflammation, and free radicals generated by oxidative stress can regulate the level of immune-related molecules, while excessive serves as the principal reason for cell injury in inflammatory disorders [27]. Cyclophosphamide can cause bone marrow suppression, which reduces the body’s antioxidant capacity. Catalase (CAT), glutathione (GSH), and superoxide dismutase (SOD) are important antioxidant enzymes. SOD has the ability to transform superoxide anion radical into hydrogen peroxide, and CAT and GSH can further decompose hydrogen peroxide through synergistic action, effectively reducing peroxide damage. Produced through lipid peroxidation, malondialdehyde (MDA) reflects the degree of oxidative stress by its presence in biological samples [28]. From Figure 6A–D, the contents of CAT, SOD, and GSH in the liver of mice from the MC group were significantly lower than those in the NC group, whereas MDA levels were markedly higher (*p* < 0.05). The BJP treatment group exhibited higher levels of CAT, SOD, and GSH in the liver than the MC group, with a concurrent decrease in MDA levels (*p* < 0.05). BJP-H exhibited no significant variance from the PC group (*p* < 0.05). In summary, BJP could reduce oxidative stress in the liver of immunocompromised mice.

#### 3.2.6. Determination of Short-Chain Fatty Acid (SCFA) Content

Findings from many studies have indicated that intestinal flora is a key component in maintaining and regulating intestinal barrier function and promoting immune function development [29]. SCFAs are primarily produced through fermentation by intestinal microbes, typically containing less than six carbon chains. The primary constituents of SCFAs are acetic acid, propionic acid, and butyric acid, which form the bulk of the total [30], and they contribute critically to the healthy function and architecture of the intestine and epithelial cells within the colon. Acetic acid, as the main energy source of intestinal epithelial cells, supports mucosal nutrient metabolism [31]. Propionic acid has an immunomodulatory function, supports intestinal barrier maintenance, and protects the mucosa by enhancing the stability of interepithelial connections [32]. Butyric acid is crucial for regulating the proliferation and apoptosis of intestinal epithelial cells and immune cells, and significantly reduces the risk of colorectal cancer by inhibiting the expression of oncogenic genes [33].

The data presented in Figure 7 indicated that the SCFA levels (acetic acid, propionic acid, and butyric acid) in the cecal contents were considerably lower in the MC group than in the NC group (*p* < 0.05). These results indicated that the immunosuppressive state induced by CTX disrupted the balance of the gut microbiome in mice, leading to lower SCFA production. The reduction in SCFA levels seen in the PC and BJP groups following intragastric administration was reversed relative to the MC group (*p* < 0.05). In addition, among the BJP dose groups, the BJP-H dose group had the best promoting effect on SCFAs, and it was statistically indistinguishable from the PC group (*p* < 0.05). The study demonstrated that BJP could stimulate the generation of short-chain fatty acids in immunosuppressed mice, leading to enhanced intestinal well-being.

## 4. Discussion

The dysfunction of the immune system may promote the development of autoimmune diseases, cancer, metabolic diseases, and other diseases [34,35]. In order to improve people’s health, the research and development of new and effective immune modulators has become increasingly critical. Polysaccharides have shown great potential as a natural source of immunomodulators [36]. In this study, by constructing a cyclophosphamide-induced immunocompromised mouse model, it was found that BJP exhibited immunomodulatory activity in immunosuppressed mice.

The structural features of polysaccharides (molecular weight, functional group, monosaccharide composition, linking mode, etc.) are closely linked to their biological activities. Molecular weight influences the conformation stability of polysaccharides. Higher molecular weight polysaccharides contain a large number of structural and functional units, thus maintaining complex spatial conformation. Molecular weight analysis revealed that RPAPS, RPAPW1, and RPAPW2 derived from *Radix Paeoniae Alba* have respective molecular masses of 1.0 × 10^5^ Da, 1.1 × 10^4^ Da, and 6.6 × 10^3^ Da. Compared to the other two polysaccharides, RPAPS with high molecular weight significantly enhanced the phagocytosis of macrophages [37,38]. Xue et al. [39] found that glucosidic bonds containing α and β types had strong biological activity. Li et al. [40] isolated two polysaccharides from *Paeonia suffruticosa Andr*, among which SAP-2, with a high content of uronic acid, had a stronger proliferative effect on RAW264.7 cells. Mannose, arabinose, and galactose were widely believed to have excellent immune-enhancing effects [41,42,43,44]. Zhao et al. [45] isolated four polysaccharides from *Plantago deoressa,* and PDSP-3 stood out for its superior ability to activate spleen cells and macrophages. It was composed of high levels of mannose and arabinose, alongside 1,3-Ara*f* and 1,3-Man*p*. In conclusion, BJP (1.42 × 10^6^ Da) may form more complex spatial conformation and more immune recognition sites. The main constituents of the monosaccharides were Rha, Ara, Gal, and Gal-UA, containing β and α-type glycosidic bonds, which not only conformed to the law of acid polysaccharide enhancing immune activity, but also had the structural basis of producing unique biological effects. Subsequent studies will further analyze the high-level structure of polysaccharide to reveal how these structures influence their immunomodulatory properties.

To study the effects of a blackening treatment on the structure and immunoreactivity of jujube polysaccharides, Liu et al. [46] prepared and purified two polysaccharides from jujube: JP (prepared by aqueous alcohol precipitation) and AC-JP (prepared by acetylation modification). Structural analysis showed that the molecular weight of JP was 2.75 × 10^5^ Da, which was mainly composed of each monosaccharide of Rha, Ara, Xyl, Man, Glu, and Gal, with a molar ratio of 105:100:161:5:107, and glyoxylate at 39.84 ± 0.58%, respectively; the molecular weight of AC-JP was 3.38 × 10^5^ Da, which was mainly composed of Rha, Ara, Xyl, Man, Glu, and Gal monosaccharides, with a molar ratio of 39:100:121:2:5:10. The study further explored the immunomodulatory effects of AC-JP by constructing an immunosuppressed mouse model. The results revealed that Ac-JP increased the thymic index and splenic index; increased white blood cell (WBC), red blood cell (RBC), and platelet (PLT) levels; and promoted the production of immune-related cytokines (including IFN-γ, IL-4, IgA, IgG, and IgM) in mice. In addition, the relative abundance of *Alloprevotella*, *Prevotellaceae_UCG-001*, and the concentration of acetic acid in the intestinal contents were increased by Ac-JP gavage. Meanwhile, the relative abundance of *Desulfovibrio* decreased. These results suggest that Ac-JP has potential as an immunomodulator. Studies have shown that higher molecular weight polysaccharides are usually associated with stronger immunomodulatory effects. Hu et al. [47] obtained five polysaccharides with different molecular weights from Juniperus communis seeds, among which the high-molecular-weight TKMSP-3 had the strongest inhibitory activity against RAW 264. 7 cells. In addition, relevant studies have shown that polysaccharides containing higher contents of arabinose, galactose, and galacturonic acid promote cell proliferation, increase the secretion of pro-inflammatory cytokines, and exhibit stronger immunostimulatory effects on macrophages [43,48,49,50]. Compared to the above-mentioned jujube polysaccharides, the blackening treatment may enhance the immunomodulatory potential of jujube polysaccharides by altering their structural characteristics, such as increasing the molecular weight, the proportion of galactose and arabinose in the composition of monosaccharides, and the content of uronic acid. This study provides a theoretical basis and experimental support for the development of blackened jujube polysaccharide as a potential natural immunomodulator. Although this study explored its immunomodulatory mechanism in vivo, further studies in metabolomics and gut flora composition are needed in the future to fully reveal the mechanism of action of its immunological activity.

## 5. Conclusions

The BJP was obtained and purified from blackened jujube as part of this investigation. Further structural characterization of BJP revealed that the main sugar residues were →3)-α-L-Ara*f-*(1→, →3,5)-α-L-Ara*f*-(1→, →3)-β-D-Gal*p*A-(1→, →2,4)-β-D-Gal*p*-(1→, →4)-β-D-Gal*p*A-(1→, →3)-α-L-Rha*p*-(1→ and →3,4)-α-L-Rha*p*-(1→. Through the construction of a cyclophosphamide-induced immunosuppression model, the findings indicated that BJP could effectively enhance the condition of immunosuppressed mice: BJP could inhibit the decline of body weight and organ index of mice, and HE staining showed that BJP had an alleviating effect on spleen and thymus tissue damage. BJP promoted the secretion of serum inflammatory factors (IFN-γ, TNF-α, IL-2, and IL-6) and immunoglobulins (IgA and IgG), and improved the liver’s oxidative stress condition (including increasing the content of SOD, CAT, and GSH, and decreasing the content of MDA). In addition, BJP promoted the production of short-chain fatty acids (including acetic acid, propionic acid, and butyric acid) in cecal contents. To conclude, the study demonstrated that polysaccharide from blackened jujube can modulate immune responses in CTX-induced immunosuppression, supporting the development of new immunomodulators.

## Figures and Tables

**Figure 1 foods-14-02531-f001:**
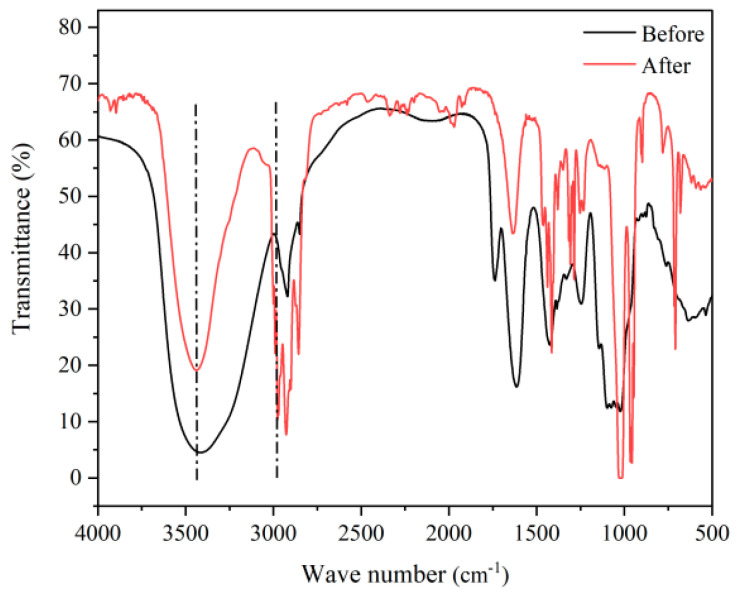
Infrared spectrum comparison of BJP before and after methylation.

**Figure 2 foods-14-02531-f002:**
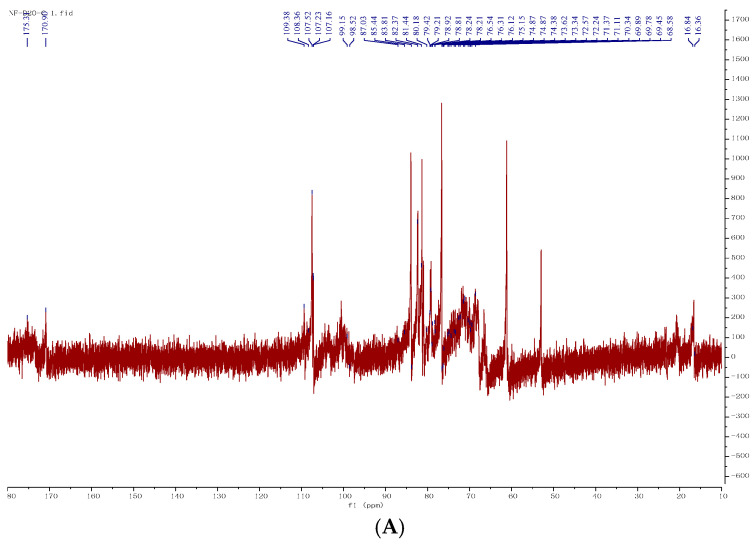
Nuclear magnetic map of BJP. ^1^H NMR (**A**), ^13^C NMR (**B**), COSY (**C**), COSY amplification (**D**), HSQC (**E**).

**Figure 3 foods-14-02531-f003:**
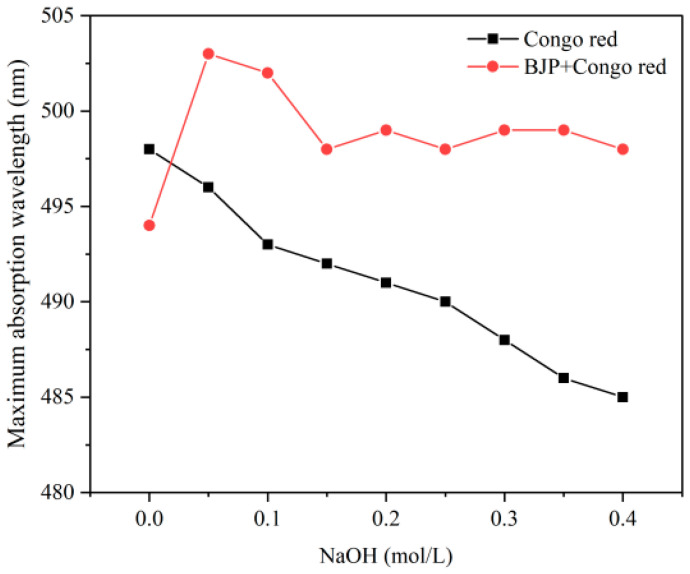
Congo red experiment.

**Figure 4 foods-14-02531-f004:**
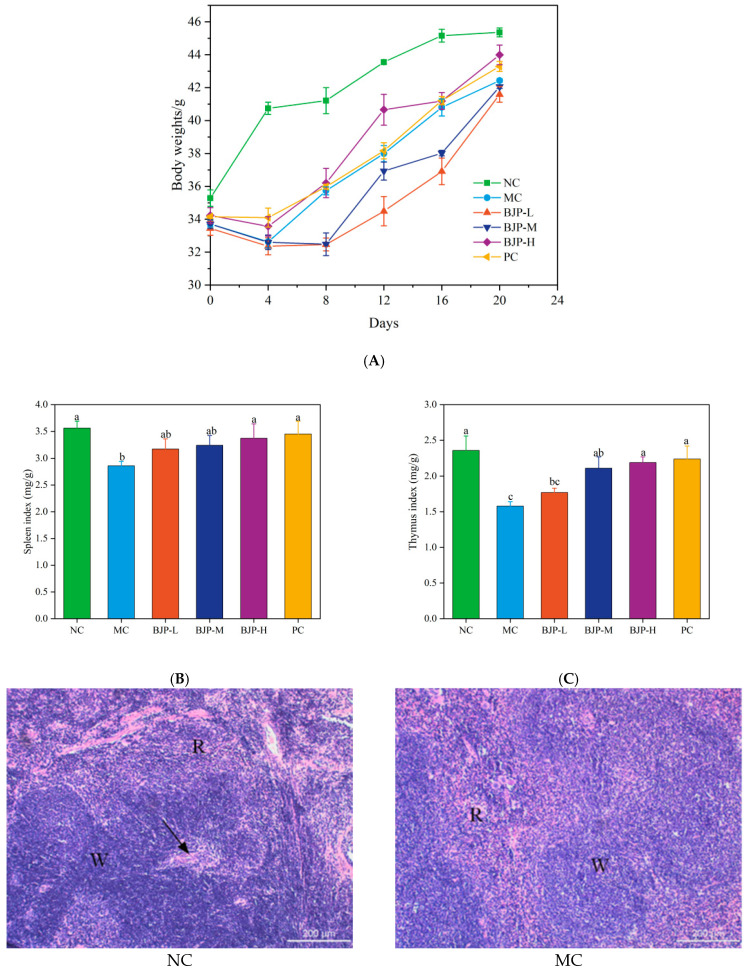
Changes in body weight of mice (**A**), spleen index (**B**), thymus index (**C**), spleen HE staining (note: R-red marrow; W—white pulp; →—splenic trabecula) (**D**), thymus HE staining (note: TM- thymus medulla; TC—thymic cortex; →—thymic corpuscles) (**E**). The data are shown as the mean ± standard deviation for each group (n = 3). Statistically significant distinctions among the groups were represented by different letters (*p* < 0.05).

**Figure 5 foods-14-02531-f005:**
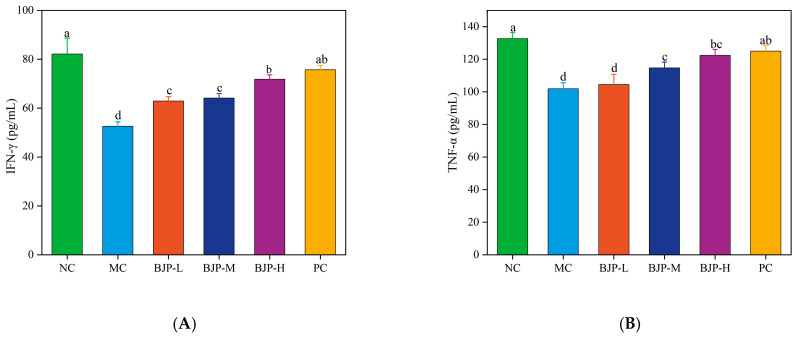
Changes in inflammatory factors and immunoglobulins in the serum of mice. IFN-γ (**A**), TNF-α (**B**), IL-2 (**C**), IL-6 (**D**), IgG (**E**), IgA (**F**). The data are shown as the mean ± standard deviation for each group (n = 3). Statistically significant distinctions among the groups were represented by different letters (*p* < 0.05).

**Figure 6 foods-14-02531-f006:**
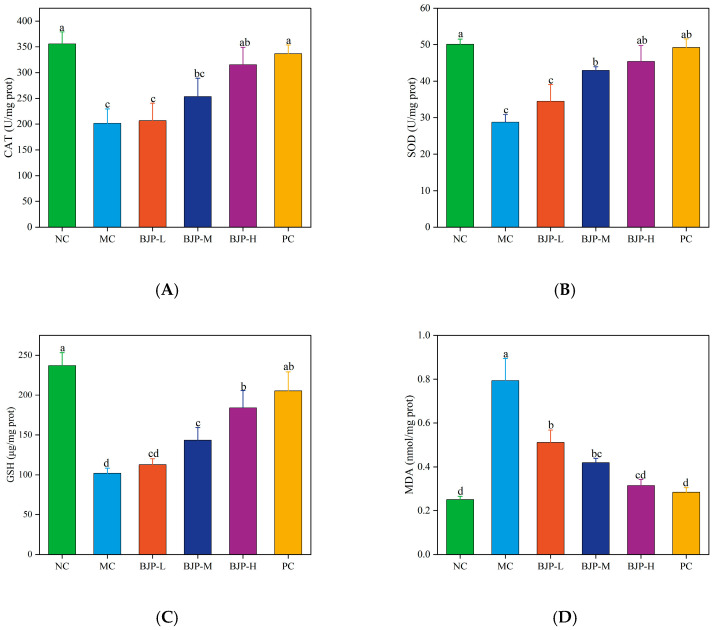
The changes in antioxidant levels in the liver of mice. CAT (**A**), SOD (**B**), GSH (**C**), MDA (**D**). The data are shown as the mean ± standard deviation for each group (n = 3). Statistically significant distinctions among the groups were represented by different letters (*p* < 0.05).

**Figure 7 foods-14-02531-f007:**
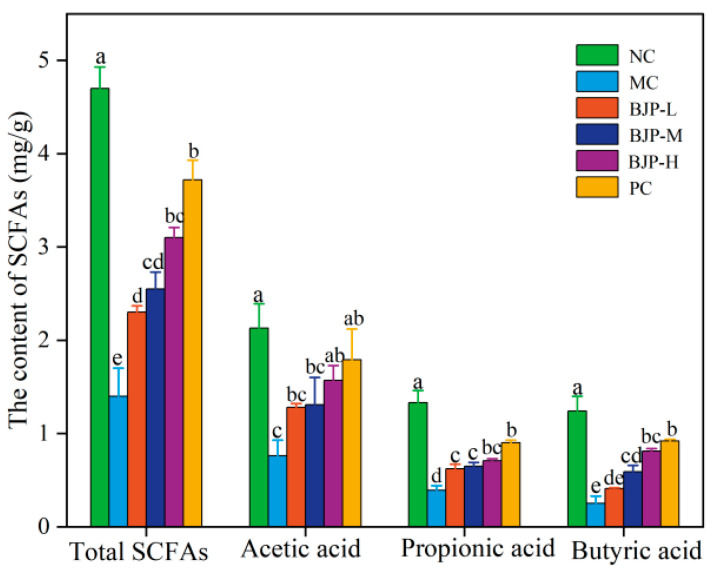
The changes in SCFAs in the cecal contents of mice. The data are shown as the mean ± standard deviation for each group (n = 3). Statistically significant distinctions among the groups were represented by different letters (*p* < 0.05).

**Table 1 foods-14-02531-t001:** Results of BJP methylation analysis.

Retention Time/min.	Methylated Sugar Residues	Connection Key Type	Main Mass Fragments (m/z)	Relative Molar Ratio
4.118	1,3,4-tri-O-Ac_3_-2,5-di-O-Me_2_-arabinitol	→3)-Ara*f*-(1→	43.1, 71.1, 85, 117, 149, 162.9	0.63
6.751	1,3,4,5-tetra-O-Ac_4_-2-O-Me-arabinitol	→3,5)-Ara*f*-(1→	57, 76.1, 93, 103.9, 131.9, 148.9, 167, 223.1, 280.8	3.56
4.118	1,3,5-tri-O-Ac_3_-2,4,6-di-O-Me_3_-galactitol	→3)-Gal*p*A-(1→	43, 72.9, 84.9, 107.9, 128.9, 163, 223.1, 281	1.08
5.021	1,2,4,5-tetra-O-Ac_4_-3,6-di-O-Me_2_-galactitol	→2,4)-Gal*p*-(1→	43.1, 74.9, 91, 102.9, 117, 130.9, 148.8, 159.1, 206.9, 280.8	1.71
7.722	1,4,5-tri-O-Ac_3_-2,3,6-di-O-Me_3_-galactitol	→4)-Gal*p*A-(1→	41, 57, 76, 104.1, 124.9, 148.9, 164.9, 223, 280.8	1.00
4.032	1,3,5-tri-O-Ac_3_-6-deoxy-2,4-di-O-Me_2_-mannitol	→3)-Rha*p*-(1→	43, 59.1, 72.9, 86, 110.9, 126.9, 148.7, 167.9, 182.8, 242.8	0.67
4.736	1,3,4,5-tetra-O-Ac_4_-6-deoxy-2-O-Me-mannitol	→3,4)-Rha*p*(1→	43.1, 60, 74.9, 87.9, 106.9, 126.9, 149, 184, 205.8	0.64

**Table 2 foods-14-02531-t002:** H and ^13^C chemical shifts in BJP.

Serial Number	Monosaccharide Residues		Chemical Shift (δ, ppm)					
			C1	C2	C3	C4	C5	C6
			H1	H2	H3	H4	H5	H6
A	→3)-α-L-Ara*f*-(1→	C	108.36	78.81	78.24	75.15	69.78	-
		H	5.38	4.56	4.12	3.91	3.81	-
B	→3,5)-α-L-Ara*f*-(1→	C	109.38	80.18	78.92	74.87	72.57	-
		H	5.13	3.68	3.51	3.77	3.60	-
C	→3)-β-D-Gal*p*A-(1→	C	107.16	83.81	76.12	74.38	70.34	170.90
		H	5.23	3.98	3.88	3.70	3.65	-
D	→2,4)-β-D-Gal*p*-(1→	C	107.23	87.03	79.21	78.21	72.24	69.89
		H	5.10	4.10	4.01	3.86	3.72	3.58
E	→4)-β-D-Gal*p*A-(1→	C	107.52	82.37	73.34	71.37	74.87	175.31
		H	5.07	4.18	3.84	3.72	3.63	-
F	→3)-α-L-Rha*p*-(1→	C	98.52	81.44	79.42	70.77	69.45	16.84
		H	5.18	4.08	3.97	3.85	3.74	1.19
G	→3,4)-α-L-Rha*p*-(1→	C	99.15	85.44	76.54	76.31	68.25	16.36
		H	5.05	4.25	4.05	3.95	3.84	1.23

Note: - indicates that it is not detected.

## Data Availability

The original contributions presented in the study are included in the article, further inquiries can be directed to the corresponding authors.

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
