# Peer review of "A Novel Polysaccharide from Blackened Jujube: Structural Characterization and Immunoactivity"

_foods, 2025, doi:10.3390/foods14142531_

Round 1

Reviewer 1 Report

Comments and Suggestions for Authors

Dear Authors,

The aim of this work is to provide useful information based on the established link between immune function and antioxidant and anti-inflammatory properties, as well as immunomodulatory effects of Blackened Jujube polysaccharides, which would be the scientific basis for their wider application. The structure of novel Blackened Jujube polysaccharides (BJP) is precisely characterized and their immunomodulatory activity is investigated, as future potential immunomodulators.

      Comprehensive tests were conducted: preparing samples for examination with “green” method of ultrasound-assisted extraction, modern GC-MS, nuclear magnetic resonance (NMR), in vivo immune activity on experimental animal (SPF Kunming male mice), histopathological analysis, ELISA test (determination of inflammatory factors in mouse serum, detection and determination of Ig in mouse serum, oxidative stress levels (CAT, SOD, GSH, MDA) in mouse liver tissue and determination of Short-chain fatty acids (SCFA) content by GC-MS.

      The topic is relevant, contemporary and up-to-date, the results contribute to a more complete understanding of the impact of the BJP on immunomodulatory activity.

      The results presented in this Manuscript (Article) provide valuable theoretical and practical insights. The methodology is modern and gives valuable results. BJP promoted the secretion of serum inflammatory factors (IFN-γ, TNF-α, IL-2, IL-6) and Ig (IgA, IgG), and improved liver's oxidative stress condition (including increasing the content of SOD, CAT, GSH, and decreasing the content of MDA). The results are comparable with the results of small number of other authors, which is shown in the chapter 3 (Results) and 4 (Discussion).

      References are the appropriate, but in all 54 references missing DOI number. The names of plant species must be in italics, both in all the cited references and in the text of the Manuscript - Article. The genus and species of bacteria must be in italics, both in all the cited references and in the text of the Manuscript - Article. For authors: proofread the entire text and references and correct any misspellings.

      One table and 7 figures (histograms; histopathological preparation - staining of thymus with H&E - hematoxylin and eosin) are very illustrative and clear.

      The reviewed study demonstrated that polysaccharide from blackened jujube can modulate immune responses in CTX-induced immuno-suppression, providing development of new immunomodulators.

      There are a number of shortcomings of a technical nature, all of which are described as comments and suggestions in the PDF document I downloaded from the SuSy platform (Manuscript). Despite the significant results and impact of the work, there is a need to make the proposed more extensive technical corrections.

Author Response

Reply to the Reviewers

Reviewer 1:

Q1: Comprehensive tests were conducted: preparing samples for examination with “green” method of ultrasound-assisted extraction, modern GC-MS, nuclear magnetic resonance (NMR), in vivo immune activity on experimental animal (SPF Kunming male mice), histopathological analysis, ELISA test (determination of inflammatory factors in mouse serum, detection and determination of Ig in mouse serum, oxidative stress levels (CAT, SOD, GSH, MDA) in mouse liver tissue and determination of Short-chain fatty acids (SCFA) content by GC-MS.

Response: We sincerely appreciate your attention to the experimental methods in the manuscript.

Q2: The topic is relevant, contemporary and up-to-date, the results contribute to a more complete understanding of the impact of the BJP on immunomodulatory activity.

Response: We sincerely appreciate your recognition of the relevance and timeliness of our research on the structure of blackened jujube polysaccharide and their immunomodulatory activity. By reviewing the available literature, we found that studies on the structural characteristics of blackened jujube polysaccharides and their biological activities are still relatively limited. Therefore, the present study aims to fill this gap and provide basic data and theoretical support for the in-depth exploration of its potential immunomodulatory effects.

Q3. The results presented in this Manuscript (Article) provide valuable theoretical and practical insights. The methodology is modern and gives valuable results. BJP promoted the secretion of serum inflammatory factors (IFN-γ, TNF-α, IL-2, IL-6) and Ig (IgA, IgG), and improved liver's oxidative stress condition (including increasing the content of SOD, CAT, GSH, and decreasing the content of MDA). The results are comparable with the results of small number of other authors, which is shown in the chapter 3 (Results) and 4 (Discussion).

Response: We are very grateful for your positive comments on our manuscript and for recognizing the theoretical and practical value of this study. We also appreciate your recognition of the modern research methods we used. As described in the paper, blackened jujube polysaccharide was able to promote the secretion of key inflammatory factors (IFN-γ, TNF-α, IL-2, IL-6) and immunoglobulins (IgA, IgG) in serum, showing their potential immunomodulatory effects. In addition, our study also showed that blackened jujube polysaccharide significantly improved hepatic oxidative stress status, as evidenced by increased levels of the antioxidant enzymes SOD, CAT, and GSH, and decreased levels of the oxidative damage marker MDA. And have compared our results with related literature in Chapter 3 (Results) and Chapter 4 (Discussion), highlighting the similarities and the unique contribution of this study.

Q4. References are the appropriate, but in all 54 references missing DOI number. The names of plant species must be in italics, both in all the cited references and in the text of the Manuscript - Article. The genus and species of bacteria must be in italics, both in all the cited references and in the text of the Manuscript - Article. For authors: proofread the entire text and references and correct any misspellings.

Response: Thank you for your professional and constructive comments. Corresponding DOI numbers have been added to all references, and the names of plants have been corrected to italics in the references and corresponding text sections.

Q5. One table and 7 figures (histograms; histopathological preparation - staining of thymus with H&E - hematoxylin and eosin) are very illustrative and clear.

Response: We sincerely appreciate your acknowledgement of the clarity and visualization of the graphs in our manuscript. We are pleased that you found the visual presentation of the data - including bar charts as well as hematoxylin-eosin (H&E)-stained histopathology sections - to be effective in supporting our findings.

Q6. The reviewed study demonstrated that polysaccharide from blackened jujube can modulate immune responses in CTX-induced immuno-suppression, providing development of new immunomodulators.

Response: We sincerely appreciate your recognition and summary of this study. Studies on the activity of blackened jujube polysaccharides are relatively limited, and our study found that blackened jujube polysaccharide exhibited immunomodulatory effects in CTX-induced immunosuppression model, demonstrating its potential as a novel immunomodulator.

Q7. There are a number of shortcomings of a technical nature, all of which are described as comments and suggestions in the PDF document I downloaded from the SuSy platform (Manuscript). Despite the significant results and impact of the work, there is a need to make the proposed more extensive technical corrections.

Response: Thank you for your professional and constructive comments. We answer the questions in the PDF document in sections as follows:

(1) Delete, unnecessary, the rest is from the Instructions for Authors!

Response: Thank you for your professional and constructive comments. The passage "keywords: keyword 1; keyword 2; keyword 3 (List three to ten pertinent keywords specific to the article yet reasonably common within the subject discipline.)" has been deleted.

(2) Device model and type, country of manufacture and address missing.  Complete, also in the rest of the text (Chapter Material and methods).

Response: Thank you for your professional and constructive comments. The equipment model, manufacturing country and address have been supplemented. The main text has been changed to Referring to the method in the literature [16], the crude polysaccharide of black jujube was extracted in three repetitions using ultrasonic extraction equipment (BILON10-250C, Shanghai Bilon Instrument Manufacturing Co, Shanghai, China).

(3) Hong et al. [17], 15 mg of ...

Response: Thank you for your professional and constructive comments. The period has been corrected to a comma.

(4). Is the NaH correct? Maybe NaOH?

Response: Thank you for your question. After consulting relevant materials, it has been confirmed that the substance should be NaH, not NaOH. Reduction of the polysaccharide chain by NaH breaks the polysaccharide chain and converts the exposed functional groups to acetyl groups by acetylation for subsequent acetylation modification reactions.

(5) Separate the degree symbol with a space from the last digit. In this way, in the text of the entire paper, all symbols and units of measurement should be separated, except for percentages, which are written next to the last digit of the number.

Response: Thank you for your professional and constructive comments. All symbols and measurement units except the percentage symbol have been written separately from the numbers, such as: 110 ℃, 100 ℃, 4 ℃.

(6) Separate the degree symbol with a space from the last digit. In this way, in the text of the entire paper, all symbols and units of measurement should be separated, except for percentages, which are written next to the last digit of the number. Also, in this place, the symbols are written in a different font than Palatino Linotype, which is used to write the entire text.

Response: Thank you for your professional and constructive comments. The format has been modified as required. All symbols and measurement units except the percentage symbol were written separately from the numbers and the font is unified as palatino linotype font, such as: 100 °C.

(7) Device model and type, country of manufacture and address missing.  Complete, also in the rest of the text (Chapter Material and methods).

Response: Thank you for your professional and constructive comments. The model and type of the equipment in the text, as well as the manufacturing country and address, have been supplemented. Such as GC-MS (GCMS7890B-7000C, Agilent Technologies Inc., USA).

(8) Enter the full name, and only then the abbreviation (in this case the method) at the first appearance of the abbreviation in the text of the Manuscript Correct everything until the end of the Manuscript, e.g. in line 142, .

Response: Thank you for your professional and constructive comments. The full name has been entered into the text and an abbreviation has been entered where the abbreviation first appears in the text such as heteronuclear single quantum coherence Spectroscopy (HSQC), and correlation spectroscopy (COSY).

(9) Device model and type, country of manufacture and address missing. Complete, also in the rest of the text (Chapter Material and methods).

Response: Thank you for your professional and constructive comments. The model and type of the equipment in the text, as well as the manufacturing country and address, have been supplemented. Such as UV spectrophotometer (UV-1600, Shanghai Meipuda Instrument Co., LTD, Shanghai, China).

(10) Correcting the sentence, trying not to place the number at the beginning of the sentence, is not in the spirit of the English language.

Response: Thank you for your professional and constructive comments. The sentence has been changed to “A total of forty-eight SPF-grade male Kunming mice, with an average body weight of 20 ± 2 g and an age of 4 ± 2 weeks, were provided by Spife (Beijing) Biotechnology Co., Ltd.”  

(11) Enter the full name before the abbreviation.

Response: Thank you for your professional and constructive comments. The full names and corresponding abbreviations of the experimental groups have been added to the text, e.g., blackened jujube polysaccharide low-dose group (BJP-L), blackened jujube polysaccharide medium-dose group (BJP-M), and blackened jujube polysaccharide high-dose group (BJP-H).

(12) Type and name of ELISA kit, country of manufacture and address missing. Complete, also in the rest of the text (Chapter Material and methods).

Response: Thank you for your professional and constructive comments. The type and name of the ELISA kit, the country of production and the address have been supplemented in the corresponding text. The serum levels of interferon-γ (IFN-γ), tumor necrosis factor-α (TNF-α), interleukin-2 (IL-2), and interleukin-6 (IL-6) in mice were measured using commercial Mouuse ELISA kits (SEKM-0031, SEKM-0034, SEKM-0004, and SEKM-0007; Beijing Solarbio Science & Technology Co., Ltd., Beijing, China).

(13) Type and name of ELISA kit, country of manufacture and address missing. Complete, also in the rest of the text (Chapter Material and methods).

Response: Thank you for your professional and constructive comments. The type and name of the ELISA kit, the country of production and the address have been supplemented in the corresponding text. The levels of immunoglobulins A and immunoglobulins G (IgA and IgG) in the serum of mice were detected using commercial Mouse ELISA kits (SEKM-0094, SEKM-0098; Beijing Solarbio Science & Technology Co., Ltd., Beijing, China).

(14) Which kit? Type and name of kit, country of manufacture and address missing. Complete, also in the rest of the text.

Response: Thank you for your professional and constructive comments. The type and name of the ELISA kit, the country of production and the address have been supplemented in the corresponding text. The levels of catalase (CAT), superoxide dismutase (SOD), reduced glutathione (GSH), and malondialdehyde (MDA) in the liver of mice were detected using commercial mouse enzyme-linked immunosorbent assay kits (BC0205, BC5165, BC1175, BC0025, Beijing Solarbio Science & Technology Co., Ltd., Beijing, China).

(15) Enter the full name before the abbreviation, at her first appearance.

Response: Thank you for your professional and constructive comments. The abbreviation "SCFAs" in the title has been changed to "Short-Chain Fatty Acids".

(16) I guess it left “undelete” in the text by mistake, Delete both in this place and in the line 218/219, and 236, 245, 246, 250, 251, 254, 255, 256, 257, 262, 273, 289/290, 300, 313, 322/323, 341, 360, 381, 399. Enter, in the square brackets the reference numbers according to the order of their apperarance, if they are missing.If it is a matter of addres...

Response: Thank you for your professional and constructive comments. An error occurred during the citation, and the cited chart has been corrected at the corresponding position.

(17) Delete "excess" spaces.

Response: Thank you for your professional and constructive comments. The redundant spaces have been deleted.

(18) Delete point, humoral immunity

Response: Thank you for your professional and constructive comments. We have removed the unnecessary punctuation and corrected the term “humoral” as suggested.

(19) Determination of Short-chain fatty acids (SCFAs) content

Response: Thank you for your professional and constructive comments. The title has been changed to "Determination of Short-chain fatty acids (SCFAs) content".

(20) Described in line 391, delete in line 394.

Response: Thank you for your professional and constructive comments. The duplicate parts in line 394 have been deleted and changed to SCFAs.

(21) The genus and species of bacteria and plants must be in italics, both in all the cited references and in the text of the Manuscript - Article. Correct wherever the genus and species of bacteria and plants are listed and are not in italics.

Response: Thank you for your professional and constructive comments. Proper nouns have been corrected to italics such as Ziziphus jujuba cv. Hamidazao, Ziziphus jujube.

(22) The genus and species of bacteria must be in italics, both in all the cited references and in the text of the Manuscript - Article. Correct wherever the genus and species of bacteria are listed and are not in italics.

Response: Thank you for your professional and constructive comments. Proper nouns have been corrected to italics such as Auricularia auricula and Auricularia polytricha Ganoderma lucidumTremella FuciformisHen-of-the-woods mushrooms (Grifola frondosa)Stolephorus chinensisMesona chinensisDictyophora rubrovalvata Radix Paeoniae Alba Paeonia suffruticosa AndrMycobacterium tuberculosisSesbania cannabinaPlantago depressaLycium barbarum.

(23) The names of plant species must be in italics, both in all the cited references and in the text of the Manuscript - Article. Proofread the entire text and references and correct any misspellings.

Response: Thank you for your professional and constructive comments. The names of plant species have been changed to italics at the corresponding positions in the main text and references. Proper nouns have been corrected to italics such as Auricularia auricula.

(24) The names of plant species must be in italics, both in all the cited references and in the text of the Manuscript - Article.

Response: Thank you for your professional and constructive comments. The names of plant species have been changed to italics at the corresponding positions in the main text and references. Proper nouns have been corrected to italics such as Lactiplantibacillus plantarum 19-2.

(25) The names of plant species must be in italics, both in all the cited references and in the text of the Manuscript - Article.

Response: Thank you for your professional and constructive comments. The names of plant species have been changed to italics at the corresponding positions in the main text and references. Proper nouns have been corrected to italics such as Bacteroides thetaiotaomicron and Faecalibacterium prausnitzii.

(26) ;

Response: Thank you for your professional and constructive comments. It has been added after the author of the literature “;”.

Reviewer 2 Report

Comments and Suggestions for Authors

The manuscript entitled „ A Novel Polysaccharide from Blackened Jujube: Structural Characterization and Immunoactivity” presented by the authors Meng Meng et al. presents the extraction, structural characterization and in vivo immunoactivity assessment of a novel polysaccharide (BJP) derived from blackened jujube. The topic is timely and relevant given the increasing interest in functional foods and natural immunomodulators. However, the manuscript requires major methodological, structural, and linguistic improvements before it can be considered for publication.

  1. Lack of consistency in the composition of BJP. Different parts of the text provide different data on the composition of BJP:

- (line 88–90) Once, the main sugar residues: Rha, Ara, Gal, GalpA are mentioned, giving only their proportions without providing possible glycosidic bonds (see below). Additionally, it would be useful to mention at least one sentence what technique was used to determine this composition.

- Another time (line 223–226), the exact bonds are given (e.g. →3,5)-α-L-Araf-(1→), →2,4)-β-D-Galp-(1→ etc.), but without information on how these proportions were determined and whether they correlate with data from HPLC, GC-MS, NMR or IR.

  1. Lack of explicit quantitative data in the main text. For example, observed repeatedly - e.g. cytokines, immunoglobulins, SCFA are presented only in the form of graphs with letters (a, b, c) without showing specific values. This makes interpretation a bit difficult and prevents meta-analysis, comparison with other works or estimation of the biological effect (effect size).
  2. In many places (e.g. lines 212, 218, 236, 273 etc.) the term "("Error! Reference source not found.")" appears in the text of the work. This is an editorial error that makes it difficult to evaluate the methods because we do not know what the data refers to (e.g. which NMR spectra, GC-MS or SCFA results are described).
  3. The “Discussion” section is underdeveloped. Rather than a critical interpretation of the findings, it reads as a summary of results already presented. No comparison with e.g. BJP-4 or other jujube polysaccharides, which also had similar immunomodulatory effects. There is no comparison with previous work (e.g., BJP-4 by Liu et al.), no biological or mechanistic explanation of the observed immunomodulatory effects, and no in-depth evaluation of the results' significance. A proper discussion should interpret results, confront them with prior findings, and identify biological mechanisms and future research directions.

Minor language errors

  1. Line 15 “This manuscript analyzed the accurately structure and immunomodulatory activity of BJP.” “Accurately” is an adverb, and the authors need an adjective: “accurate. Correct form: “...analyzed the accurate structure...”
  2. Line 419 “BJP had immunomodulatory activity on immunosuppressed mice.” A better form would be: “BJP exhibited immunomodulatory activity in immunosuppressed mice.”

Author Response

Reviewer 2:

The manuscript entitled „ A Novel Polysaccharide from Blackened Jujube: Structural Characterization and Immunoactivity” presented by the authors Meng Meng et al. presents the extraction, structural characterization and in vivo immunoactivity assessment of a novel polysaccharide (BJP) derived from blackened jujube. The topic is timely and relevant given the increasing interest in functional foods and natural immunomodulators. However, the manuscript requires major methodological, structural, and linguistic improvements before it can be considered for publication.

Q1 1.Lack of consistency in the composition of BJP. Different parts of the text provide different data on the composition of BJP:

- (line 88–90) Once, the main sugar residues: Rha, Ara, Gal, GalpA are mentioned, giving only their proportions without providing possible glycosidic bonds (see below). Additionally, it would be useful to mention at least one sentence what technique was used to determine this composition.

- Another time (line 223226), the exact bonds are given (e.g. 3,5)-α-L-Araf-(1), 2,4)-β-D-Galp-(1 etc.), but without information on how these proportions were determined and whether they correlate with data from HPLC, GC-MS, NMR or IR.

Response: Thank you for your professional and constructive comments. The methodology for the determination of monosaccharide composition has been added to the text, and the monosaccharide composition of extracted and purified blackened jujube polysaccharide (BJP) was analyzed by ion chromatography in a previous experiment (revised at lines 88-91). In this study, we continued the in-depth study of the structure of blackened jujube polysaccharide by analyzing the glycosidic bond linkage types by methylation of blackened jujube polysaccharide. The blackened jujube polysaccharide was subjected to the reaction processes of methylation, degradation and reduction, and acetylation, and the methylated PMAAS was dissolved in dichloromethane, filtered through organic membranes, and then subjected to GC-MS analysis and compared with the database, which led to the determination of the types of methylated sugar residues (modified at lines 240-246). The purified blackened jujube polysaccharide was identified as homogeneous polysaccharide by high performance liquid chromatography for subsequent structural analysis. The infrared spectra indicated that the blackened jujube polysaccharide (BJP) was an acidic sugar this was consistent with the high content of glyoxalate in the results of the monosaccharide composition. The type and molar ratio of methylated sugar residues were consistent with the major monosaccharides in the monosaccharide composition. The conformation and linkage of glycosidic bonds were further determined based on 1HNMR, 13CNMR, COSY and HSQC spectra in NMR spectroscopy (modified and added at lines 251-252).

Q2. Lack of explicit quantitative data in the main text. For example, observed repeatedly - e.g. cytokines, immunoglobulins, SCFA are presented only in the form of graphs with letters (a, b, c) without showing specific values. This makes interpretation a bit difficult and prevents meta-analysis, comparison with other works or estimation of the biological effect (effect size).

Response: We appreciate your attention to the way quantitative data are presented in the main text. In recent years, it has become common for researchers in the field to depict key bioactivity indicators in the form of trends and graphs, especially when the focus is on biological significance rather than precise numerical differences:

  1. Son et al. in Immunostimulating effects of ulvan type polysaccharide isolated from Korean Ulva pertusa in cyclophosphamide-induced immunosuppressed BALB /c mice article explored the immunoreactive effects of polysaccharide (UPP) by constructing an immunocompromised mouse model, and found that cytokines (IL-6, IL-12, and TNF-α) in serum were significantly reduced in the NC group. Krestin (PC group) treatment showed a slight but not statistically significant increased in serum cytokine levels compared to the NC group. However, UPP significantly increased the levels of all inflammatory cytokines measured in serum. levels of IL-6 (p < 0.05) and IL-12 (p < 0.05) were significantly higher in the UPP-H group compared to the PC group. In addition, UPP administration had a stimulatory effect on cytokines in serum and spleen tissues. The results for immunoglobulin A were consistent with those for cytokines in serum and spleen tissues. In contrast, the determination of short-chain fatty acids was described as a reduction in the levels of acetic, propionic, and butyric acids to half that of the NT group after CTX treatment in the NC group (p < 0.001). Prophylactic administration of Krestin (PC group) had an incomplete preventive effect on the three SCFAs.UPP-L and UPP-M had no significant effect on acetic acid production, but the UPP-H group had a similar effect to the NT group (p < 0.001). In addition, all experimental groups receiving UPP showed propionic and butyric acid production. In particular, the difference between the UPP-H group and the Krestin group was statistically significant (p < 0.05). In conclusion, total SCFA loss due to immunosuppression was prevented in the UPP-H group.
  2. Meng et al. in A polysaccharide from Pleurotus citrinopileatus mycelia enhances the immune response in cyclophosphamide-induced immunosuppressed mice via p62/Keap1/Nrf2 signal transduction pathway article to explore the therapeutic effect of polysaccharide (CMP) on cyclophosphamide-induced immunosuppressed mice. Compared with the normal group, the levels of IFN-γ, IL-12 and TNF-α were significantly higher in the splenic tissues in the model group (p < 0.05).The expression levels of IL-4 and IL-10 were significantly lower (P < 0.05). Compared with the model group, the expression levels of TNF-α, IFN-γ, IL-4 and IL-10 mRNA were reduced in the CMP and levamisole groups, and the difference was statistically significant (p < 0.05). The results suggested that CMP significantly decreased the expression of Th 1 cytokines and increased the expression of Th 2 cytokines in the spleen tissues of CTX-induced mice. The immunoglobulin (IgA, IgG and IgM) levels in serum and spleen were significantly lower in the model group than in the normal group (p < 0.05). Compared with the model group, the levels of IgA, IgG, and IgM were significantly higher in the M-CMP, H-CMP, and levamisole groups (p < 0.05). Thus, CMP improved CTX-induced low immunoglobulin levels. The levels of SOD, CAT and GSH-Px in the model group were found to be higher than those in the control group in the determination of antioxidant function of serum and spleen in mice (p < 0.05). The activities of serum SOD, CAT and GSH-Px were also measured in all groups of rats, and the differences between the model group and the control group were statistically significant (p < 0.05 and p < 0.01, respectively).The indices of SOD, CAT and GSH-Px in the M-CMP group, the H-CMP group and the levamisole group were all significantly elevated (p < 0.05). The MDA index of mice in the model group was significantly higher compared with the normal group (p < 0.05). Compared with the model group, the MDA indices of both CMP and levamisole groups were reduced to different degrees (p < 0.05).The difference between the H-CMP group and levamisole group was not statistically significant (p > 0.05). It indicated that CMP could improve CTX-induced spleen and serum antioxidant damage.

Referring to the relevant literature, in the original manuscript we adopted a generalized way of describing the different activity indicators, using significance markers (a, b, c) to indicate statistically significant differences between groups (p < 0.05), and the key results (e.g., cytokine levels, immunoglobulin concentrations, and SCFA distributions) were mainly presented in graphical form, with a focus on showing the trend of the changes between the different experimental groups. We believe that this method can effectively convey the biological significance of the study results, while readers can also roughly judge the changes of the data through the graphs.

Q3. In many places (e.g. lines 212, 218, 236, 273 etc.) the term "("Error! Reference source not found.")" appears in the text of the work. This is an editorial error that makes it difficult to evaluate the methods because we do not know what the data refers to (e.g. which NMR spectra, GC-MS or SCFA results are described).

Response: Thank you for your professional and constructive comments. An error was made in the citation process and the charts and tables cited in the text have been corrected at the appropriate places (lines 232, 238, 258, 267, 271/272, 274, 276, 277, 281, 292, 308, 318, 331/332, 341, 360, 379, 400, 418).

Q4. The “Discussion” section is underdeveloped. Rather than a critical interpretation of the findings, it reads as a summary of results already presented. No comparison with e.g. BJP-4 or other jujube polysaccharides, which also had similar immunomodulatory effects. There is no comparison with previous work (e.g., BJP-4 by Liu et al.), no biological or mechanistic explanation of the observed immunomodulatory effects, and no in-depth evaluation of the results' significance. A proper discussion should interpret results, confront them with prior findings, and identify biological mechanisms and future research directions.

Response: Thank you for your professional and constructive comments. In the Discussion section, the literature on the immunological activity of jujube polysaccharide was cited, and it was found that jujube polysaccharide had immunological activity through the construction of the cyclophosphamide-immunocompromised mouse model, and that the blackening treatment caused structural changes to the jujube polysaccharide, which in turn may have led to the enhancement of its immunological activity. The modifications were specified below: Effect of blackening treatment on the structure and immunoreactivity of jujube polysaccharides. Liu et al [46] prepared and purified two polysaccharides from jujube: JP (prepared by aqueous alcohol precipitation) and AC-JP (prepared by acetylation modification). Structural analysis showed that the molecular weight of JP was 2.75×105 Da, which was mainly composed of each monosaccharide of Rha, Ara, Xyl, Man, Glu, and Gal, with a molar ratio of 105:100:161:5:107, and glyoxylate of 39.84±0.58%, re-spectively; the molecular weight of AC-JP was 3.38×105 Da, which was mainly com-posed of Rha, Ara, Xyl, Man, Glu, and Gal monosaccharides, with a molar ratio of 39:100:121:2:5:10. The study further explored the immunomodulatory effects of AC-JP by constructing an immunosuppressed mouse model. The results revealed that Ac-JP increased thymic index and splenic index, increased white blood cell (WBC), red blood cell (RBC), and platelet (PLT) levels, and promoted the production of immune-related cytokines (including IFN-γ, IL-4, IgA, IgG, and IgM) in mice. In addition, the relative abundance of Alloprevotella, Prevotellaceae_UCG-001, and the concentration of acetic acid in the intestinal contents were increased by Ac-JP gavage. Meanwhile, the relative abundance of Desulfovibrio decreased. These results suggest that Ac-JP has potential as an immunomodulator. Studies have shown that higher molecular weight polysaccha-rides are usually associated with stronger immunomodulatory effects. Hu et al [47] obtained five polysaccharides with different molecular weights from Juniperus com-munis seeds, among which the high molecular weight TKMSP-3 had the strongest in-hibitory activity against RAW 264. 7 cells. In addition, relevant studies have shown that polysaccharides containing higher contents of arabinose, galactose, and galacturonic acid promote cell proliferation, increase the secretion of pro-inflammatory cytokines, and exhibit stronger immunostimulatory effects on macrophages [48-51]. Compared with the above-mentioned jujube polysaccharides, the blackening treatment may en-hance the immunomodulatory potential of jujube polysaccharides by altering their structural characteristics such as increasing the molecular weight, increasing the composition of galactose and arabinose in the composition of monosaccharides, and the content of uronic acid. This study provides theoretical basis and experimental support for the development of blackened jujube polysaccharide as a potential natural immunomodulator. Although this study explored its immunomodulatory mechanism in vivo, further studies in metabolomics and gut flora composition are needed in the future to fully reveal the mechanism of action of its immunological activity.

Minor language errors

Q5. Line 15 “This manuscript analyzed the accurately structure and immunomodulatory activity of BJP.” “Accurately” is an adverb, and the authors need an adjective: “accurate. Correct form: “...analyzed the accurate structure...”

Response: Thank you for your professional and constructive comments. In line 15, the adverb “accurately” has been replaced with the adjective “accurate”.

Q6. Line 419 “BJP had immunomodulatory activity on immunosuppressed mice.” A better form would be: “BJP exhibited immunomodulatory activity in immunosuppressed mice.”

Response: Thank you for your professional and constructive comments. The sentence “BJP had immunomodulatory activity on immunosuppressed mice.” in the text has been replaced by “BJP exhibited immunomodulatory activity in immunosuppressed mice.”.
